# Mechanisms of Food-Induced Symptom Induction and Dietary Management in Functional Dyspepsia

**DOI:** 10.3390/nu13041109

**Published:** 2021-03-28

**Authors:** Kerith Duncanson, Grace Burns, Jennifer Pryor, Simon Keely, Nicholas J. Talley

**Affiliations:** 1College of Health, Medicine and Wellbeing, University of Newcastle, Callaghan, NSW 2308, Australia; g.burns@newcastle.edu.au (G.B.); jennifer.pryor@uon.edu.au (J.P.); simon.keely@newcastle.edu.au (S.K.); 2Centre for Research Excellence, Digestive Health, Hunter Medical Research Institute, New Lambton Heights, NSW 2305, Australia; 3Department of Gastroenterology, John Hunter Hospital, New Lambton Heights, NSW 2305, Australia

**Keywords:** functional dyspepsia, dietary management, gastrointestinal symptoms

## Abstract

Functional dyspepsia (FD) is a common disorder of gut-brain interaction, characterised by upper gastrointestinal symptom profiles that differentiate FD from the irritable bowel syndrome (IBS), although the two conditions often co-exist. Despite food and eating being implicated in FD symptom induction, evidence-based guidance for dietetic management of FD is limited. The aim of this narrative review is to collate the possible mechanisms for eating-induced and food-related symptoms of FD for stratification of dietetic management. Specific carbohydrates, proteins and fats, or foods high in these macronutrients have all been reported as influencing FD symptom induction, with removal of ‘trigger’ foods or nutrients shown to alleviate symptoms. Food additives and natural food chemicals have also been implicated, but there is a lack of convincing evidence. Emerging evidence suggests the gastrointestinal microbiota is the primary interface between food and symptom induction in FD, and is therefore a research direction that warrants substantial attention. Objective markers of FD, along with more sensitive and specific dietary assessment tools will contribute to progressing towards evidence-based dietetic management of FD.

## 1. Introduction

More than one in ten Australians have chronic or relapsing unexplained upper gastrointestinal (GI) symptoms, half of whom have symptoms severe enough to require a general practitioner (GP) consultation for diagnostic and therapeutic purposes [1]. In some cases further investigations (e.g., gastroduodenal biopsy, 24 h esophageal pH testing) for structural disease may identify explanations for the symptoms, including *H. pylori* gastritis, gastro-oesophageal reflux disease (GORD), coeliac disease or eosinophilic gastroenteritis [2]. However, the vast majority who present with these symptoms have a normal gastroscopy with no evidence of peptic ulceration or gastric cancer and fulfill the diagnostic criteria for functional dyspepsia (FD) using the ROME IV criteria.

Functional dyspepsia is one of the commonest disorders of gut-brain interaction, previously termed functional gastrointestinal disorders (FGIDs), FD is further categorized into epigastric pain syndrome (EPS), or eating-related post-prandial distress syndrome (PDS) [2]. These upper GI symptom profiles differentiate FD from the irritable bowel syndrome (IBS), which is characterised by lower abdominal pain and bowel dysfunction (and often bloating). [3]. By definition FD or IBS cannot be diagnosed with routine or specific diagnostic tests because there are no established structural or biochemical pathologies, but this paradigm is likely to be simplistic and sub-clinical pathologies (e.g., eosinophilic duodenitis, mast cell activation) have recently been identified that may explain symptoms in at least some subsets [4,5]. To further complicate diagnoses and management, patients are often afflicted with both IBS and FD, and these conditions also overlap significantly with gastro-oesophageal reflux disease (GORD) [6,7].

The combination of its functions and upper GI location have resulted in the duodenum being increasingly implicated in FD pathogenesis. The duodenum receives partially digested food in chyme from the stomach, and the microvilli on its absorptive enterocytes uptake water, nutrients and vitamins. The critical and unique digestive and homeostatic roles of the duodenum include neutralising the acidic chyme; maintaining the mucous-bicarbonate barrier to protect the epithelium from enzymatic damage and sensitisation to food antigens [8,9,10]; releasing gastric hormones; moderating gastric and pancreatic secretions; moderating gastric emptying and satiety [10,11,12]; moderating host-microbiota interactions; and regulating adaptive immune responses along the gastrointestinal mucosal surface [13]. The duodenal microbiota is crucial in supporting small intestinal digestive functions by fermenting food components and releasing digestive enzymes not otherwise produced by the host. This is key as appropriate digestion of dietary proteins is necessary to prevent inappropriate immune activation towards foods [14,15]. When factors alter or deplete the microbiota, for example GI infection or excess antibiotic use, this may result in a state of microbial dysbiosis where GI symptoms may be heightened.

Pathological findings in FD include increased peripheral TNF-α, IL-β and gut homing T cells, and duodenal eosinophilia [16]. Mast cells and eosinophils close to submucosal plexus neurons have been observed in this population, along with altered neuronal responsiveness [17,18,19].

Despite FD symptoms often being associated with eating and FD being as prevalent and debilitating as irritable bowel syndrome (IBS), there is no evidence-based, food-specific hypothesis for FD aetiology and dietary management approaches in FD remain largely undescribed. The efficacy of a low fermentable oligosaccharide, disaccharide, monosaccharide and polyol (FODMAP) diet in FD dietary management does not have the strong evidence base that is apparent for IBS [20]. GP and primary care dietitian awareness of FD diagnostic criteria and referral pathways have not been investigated, but are purported to be low. Given that specialist dietetic services for IBS dietary management are limited, it is likely that equivalent services for FD are even more restricted. Although the symptoms of FD are associated with eating, FD aetiology and pathophysiology are highly heterogenous, as are the foods and nutrients reported to induce symptoms [21]. As a result, people with FD have frequent health care consultations and high utilisation of pathology and endoscopy, with a subsequent very high cost burden [22]. Improved understanding of FD aetiology and pathophysiology is needed to inform clear diagnostic and referral pathways. In parallel, evidence-based dietary management approaches in primary care that parallel IBS dietary management are also needed.

A 2015 survey of gastroenterologists in the USA revealed that 90% of respondents (*n* = 1949) felt that diet therapies were at least as good or superior to existing pharmacotherapies for IBS, but FD was not assessed. A low FODMAP diet is effective for symptom management in 70% of people diagnosed with IBS, with psychological therapy and specific complementary therapies also reported to assist in symptom management [23]. Dietary guidelines for IBS management are available to guide GPs in IBS management and dietetic referral processes [24,25]. A recent study reported that service reorientation towards a ‘dietitian-first’ gastroenterology clinic model in Australia for people with gastrointestinal symptoms (and no ‘red flags’ for structural disease) led to low re-referral rates up to 24 months post-discharge and lower health service usage compared to people who consulted through the traditional model of care [26]. Together, these findings indicate that dietary intervention is a cornerstone strategy in IBS management.

FD is a disorder of symptoms and subtle immune changes, so the overarching goal of dietetic management is to complement medical therapy by using dietary modification to alleviate symptoms. Currently, guidance for dietetic management of FD is limited to a focus on frequent small meals, and possible trialing of reduced dietary fat intake to ameliorate slow gastric motility [27]. Due to the multifaceted nature of FD and broad range of presentation scenarios for people seeking dietary management advice for the condition, we believe that a differential dietary management approach for FD is needed. This model would encompass the presentation history, primary symptoms, possible aetiology and pathophysiology be considered in formulation of a staged exclusion diet. The aim of this narrative review is to collate the possible mechanisms for eating-induced and food-related symptoms and use the available evidence to develop and present a staged process for dietary management of FD.

## 2. Functional Dyspepsia Presentation and Shared Care

### 2.1. Presenting Symptoms and Medical History Related to Dietary Management of Functional Dyspepsia

The dietary assessment and management approach in FD relies on symptom history, past treatment outcomes, FGID diagnostic investigation results and related medical history. A comprehensive FD dietary management plan will be influenced by a wide range of related factors, including whether a person with FD has a history of GI infections or traveller’s diarrhoea, regular or repeated antibiotic use, intake of non-steroid anti-inflammatory drugs or stress-related or early life factors, smoking and weight status [28].

Predisposing and risk factors for FD should inform the dietetic management approach, prioritisation and potentially subsequent referral for additional testing or treatment. A specific example of this might be scenario of a food or water borne infection resulting in persistent FD symptoms. People with post infectious gastroenteritis have 2.5 times higher odds of FD at six months post infection compared to uninfected people [29]. Therefore, travel and food contamination history have implications for FD, plus testing for parasitic infection (e.g., giardia) may be warranted before commencing dietary intervention. The medical and medication history may also provide a pathway or prioritisation structure for initial dietary management approaches by alerting the dietitian to possible dysbiosis, inflammatory or gut-brain factors.

### 2.2. Availability, Access and Referral to Specialist FD Dietetic Services

Dietitians are qualified and trained to ensure nutritional adequacy while modifying dietary intake to meet client’s health and medical needs, and are therefore well positioned as the primary providers of dietary assessment, advice and management in FGIDs. The GP or gastroenterologist (GE) have crucial medical diagnostic, medical testing and referral roles, as well as providing evidence-based advice to address dietary concerns of patients. A patient with a FGID may also have a psychologist or other allied health professionals involved in their care. A shared care approach to FD management is particularly important because of the multi-factorial nature of the condition. Access to the patients’ complete related medical history and past treatment approaches is highly instructive for the treating medical and allied health team, and helps to prevent unnecessary duplication of medical history reporting or testing [30].

Referral practices of primary care GPs or GEs to dietitians for FD dietetic management have not been reported. In a study about dietary management of IBS by GEs, Lenhart et al. (2018) reported that more than half of the 1500 GEs surveyed were ‘comfortable’ or ‘very comfortable’ providing dietary counselling for IBS, with the most common modalities being provision of handouts (81%), referral to a dietitian (70%) and verbal advice (52%) [23]. Paradoxically, only 21% of GEs reported referring to a dietitian ‘usually’ or ‘almost always’. Of these referrals, 50% were to general dietitians and 30% to specialist GI dietitians, despite the preference of GEs for specialist dietetic referral options [31].

In a study involving 80 GE patients who reported trialing the low FODMAP diet for IBS, the diet was recommended by the GE in 53%, by a GP in 22% and by a dietitian for 9% of the patients. [32]. Thirty percent consulted a dietitian for low FODMAP guidance and 55% reported at least 50% symptom improvement, despite many not reaching the therapeutic FODMAP intake target. Dietetic education improved achievement of therapeutic FODMAP intake, and compliance with food challenges and maintenance phases of the FODMAP diet [32].

The need for improved communication about gastroenterology service expectations is highlighted in a 2009 study of 21 patient/GE pairs to assess alignment of specialist and referred patient’s expectations and perspectives about diagnosis and management. Almost half of the GEs in the study underestimated patients’ perception of symptom burden with 43% overestimating patients’ ability to cope in the longer-term without a desired diagnosis or specific treatment regimen. GEs were accurate in gauging that patients beliefs that diet and stress were prime symptom causes, but focused consultation more on symptom control and medication than on dietary modification [33]. It was not reported whether this approach related more to scope of practice and subsequent referral to a dietitian, or other factors.

Although not reported in the literature, it is highly likely that a lower proportion of people with FD compared to IBS are referred to a dietitian, given that there is not an equivalent evidence-based approach to the low FODMAP diet for IBS. Dietitians who specialise in FGIDs therefore need to have well-developed dietetic advocacy skills to ensure that access to dietetic services is optimised. GPs, GEs, non-specialist dietitians and other allied and primary health care clinicians are all potential referral sources. In this review, we describe and delineate the respective roles of the dietitian, GP and GE in dietary aspects of FD management, and provide guidance for dietitians on prioritisation and implementation of specific dietary management approaches.

## 3. Relationship between FD and Eating

The biopsychosocial model that distinguishes functional gastrointestinal disorders from organic gastrointestinal conditions was conceived in the late 1980s, with associated ‘Rome Criteria’ developed to differentiate between symptom groups [34]. The criteria for FD diagnosis has evolved over four iterations of the Rome Criteria to now be classified as PDS or EPS, with symptom induction after eating therefore being a key feature that differentiates PDS from EPS [4].

### 3.1. Does Eating Induce or Relieve FD Symptoms?

Bothersome post-prandial fullness at least three time a week and early satiation are the defining characteristics of PDS FD sub-type [4]. These dominant symptoms may be compounded by post-prandial epigastric pain, burning, bloating, belching and nausea. If a person with FD reports that eating makes symptoms worse (or better) or if the frequency or meal size affect symptoms, then altering meal frequency or volume in a short term (weeks not months) dietary trial is a logical first line dietary management approach. If specific foods or nutrients are reported to exacerbate symptoms, this will further inform initial dietary advice and influence subsequent management.

If epigastric pain or burning is experienced at least one a week, but not necessarily associated with eating, the medical professional may have diagnosed EPS [2]. Modifying meal size and frequency would not be a priority dietary management strategy for people with the EPS subtype of FD, although other types of dietary modifications may be indicated or warranted (see Figure 1).

For either PDS or EPS to be diagnosed, the GP will often have excluded (or trialed treatment for) GORD. At this early stage of dietary management, it is useful to the dietitian to know whether the person with FD has been prescribed a proton pump inhibitor (PPI) for their presenting symptoms and if so, how effective it has been in symptom management and how long it has been used. PPIs have been reported as beneficial in symptom reduction amongst patients with FD [35]. This presentation type may then be classified as PPI-responsive FD. The efficacy of PPIs in FD, especially for PDS symptom management, may be due to anti-inflammatory actions [36] and lessening duodenal eosinophilia [37]. Wauters et al. (2020) demonstrated that anti-eosinophil effects of short term PPI therapy, not acid suppression or barrier protection, likely reduced upper GI symptoms in treatment responsive FD. In the same study, duodenal mucosal inflammation, mucosal hyperpermeability, luminal and systemic changes were reported in FD compared to healthy controls [35].

As part of FD dietary management planning, dietitians should be aware that PPI use is associated with decreased bacterial diversity and a shift towards Streptococci species, a profile typical of the oral microbiota [38]. Dietary management strategies related to bacterial diversity and abundance are described in detail in Section 7.

### 3.2. Are Symptoms or Test Results Consistent with Delaying Gastric Emptying or Impaired Gastric Accommodation?

FD symptoms overlap substantially with gastric motor dysfunction symptoms of upper abdominal pain, cramps, fullness, satiety, nausea, and vomiting [19]. Delayed gastric emptying in FD may be associated with increased immune cell activation, increased duodenal permeability [19] or low grade inflammation via dysregulation of the neuroimmune system disturbing GI motility and visceral sensitivity [16,17,18].

An individual’s dietary intake is unlikely to be the underlying cause of delayed gastric emptying or accompanying FD symptom induction, however protein, fats, carbohydrate, alcohol and fibre are all reported to influence gastric emptying or accommodation. Lipid is considered as the most potent nutrient modulator of gut motility, regulating gastric distention via gut hormones, particularly cholecystokinin (CCK) [39]. Dietary fats have consistently been associated with FD symptom induction in cross-sectional and intervention studies [21]. CCK released from enteroendocrine cells also acts on local vagal sensory fibers in response to dietary protein, affecting gastric motility via a vago-vagal loop and also stimulating satiety through low-affinity vagal CCK receptors that signal the brain [40]. Elevated concentrations of protein and amino acids may also be detected by the hypothalamus and trigger metabolic signaling to slow gastric motility [40].

Replacing fats or protein with carbohydrate generally enhances upper gastric motility, but the motility response may also vary between types of carbohydrates [41]. Although gastric motility responses to different fermentable carbohydrates is yet to be investigated in FD, IBS sufferers report higher gastrointestinal symptom scores after fructan infusion than healthy controls [41].

Dietary glycaemic index and glycaemic load would also be expected to affect gastrointestinal symptoms, being directly related to the rate that carbohydrate is absorbed and the extent of impact on blood glucose levels. A diet with a high glycaemic index or glycaemic load preferentially enhances some bacterial strains and increases cortisol and insulin secretion, which may further compromise gastric motility [42]. A cross sectional study involving 2987 adults was used to investigate differential responses to glycaemic index and glycaemic index on gastric motility in FD. Dietary glycaemic index and glycaemic load were estimated using a validated food-frequency questionnaire and uninvestigated chronic dyspepsia was determined using Rome III criteria. After controlling for confounders and stratifying analysis by sex and body mass index (BMI), respectively, high glycaemic load was associated with an increased risk of uninvestigated chronic dyspepsia in men (OR = 2.14; 95% CI: 1.04, 4.37; *p* = 0.04) and in healthy weight adults (OR = 1.78; 95% CI: 1.05, 3.01; *p* = 0.03). These data suggest that there are BMI and sex-specific associations between dietary carbohydrate quality with functional dyspepsia [42].

Dietary management of delayed gastric emptying is challenging. A proportion of people who present to a dietitian for FGID management have completed a gastric motility test for suspected gastroparesis or report primary symptoms that are consistent with delaying gastric emptying or impaired gastric accommodation. For this subset, the initial dietary strategies for dietitians to consider would include modifying meal size, frequency, texture or nutrient composition, and manage co-morbidities such as diabetes mellitus (ensuring tight glycemic control) [43]. Gastric motility may be improved with smaller meals, finer texture (less mechanical processing), lower fat and protein, separating solids and liquids (handled differently by the stomach) and higher glycaemic index food combinations. Nutrient specific dietary advice for symptoms beyond slow gastric motility are outlined in more detail in the following sections.

## 4. Nutrient-Specific Dietary Management of FD

### 4.1. Are Epigastric Symptoms Attributable to Specific Macronutrient/s?

The general public are becoming increasingly aware of the nutrient composition of foods, to the extent that it is common for foods like bread and pasta to be referred to as ‘carbs’ and meat and dairy to be labelled ‘protein’, based on the perceived dominant nutrient. A high proportion of people with FD report food ingestion as inducing symptoms and alleviation of symptoms with dietary modification [21]. Dietary hypervigilance can be a ‘double edged sword’ for dietetic management of people with FD, who are likely to be acutely aware of their own dietary intake, but may have developed misconceptions or been misinformed about the nature of food-nutrient interactions. Nevertheless, this heightened dietary awareness means that FD sufferers often maintain a detailed food history, which can be useful in preliminary dietetic management in forming ‘likely suspects’ for dietary intervention prioritization. All four energy-providing macronutrients: fat; carbohydrate; protein; and alcohol, are reported as contributing to GI symptoms in FD [44], however most studies investigating associations between diet and FD have been cross sectional so causation remains uncertain.

### 4.2. Do Specific Carbohydrates Induce FD Symptoms or Does Removal of Specific Carbohydrates Alleviate Symptoms?

Carbohydrates are present in most foods, but are more highly concentrated in grains, vegetables, fruits, legumes and as sugars in discretionary foods [45]. Carbohydrates are classified by chain length: short-chain carbohydrates contain chains of up to 10 sugars, and longer chains with more complex linkages are classified as oligosaccharides. The low FODMAP diet for IBS management is an evidence-based dietary management approach that involves short term reduction and sequential reintroduction of specific fermentable carbohydrates [46]. Randomised-controlled trials, observational and comparative studies indicate that a low FODMAP diet offers considerable symptom relief in 50 to 80% of people with IBS [47]. Long term FODMAP restriction is not recommended due to potential dysbiosis [48]. Ideally, an individual with IBS would achieve a threshold for FODMAP tolerance and support this with lifestyle management strategies.

The low FODMAP principle of excluding carbohydrates that are not fully absorbed in the small intestine is potentially applicable to FD management. It is feasible that FODMAP carbohydrates exert osmotic effects in the intestinal lumen to increase water volume and that some carbohydrate FODMAPs are fermented by small intestinal bacteria, resulting in gas production [46]. People with FD may experience symptoms due to visceral hypersensitivity to either osmotic load, gas production or both, but in the epigastric region rather than the large bowel. The low FODMAP diet has been proposed as a potential FD dietary management approach in South East Asia, although cultural factors around reporting of bowel symptoms indicate a high potential for misdiagnosis of IBS as FD in Asian populations [20].

Wheat is a fibre- and carbohydrate-rich grain that seems particularly problematic for people with FD. Potter et al. (2018) reported a significant association between self-reported wheat sensitivity and FD [49], which led to a hypothesis that a major subgroup of functional dyspepsia is induced by wheat. Wheat is high in both fructan FODMAPs and in immune-response inducing proteins such as gliadin (see section below on protein), so differentiating which wheat components induce which FD symptoms should be a research priority.

Undigested carbohydrate that is available to gut bacteria for fermentation to short chain fatty acids (SCFA) is classified as fibre. The carbohydrate chain length and water solubility determine the fermentability of fibres, ranging from highly fermentable short chain fructo-oligosaccharides and galacto-oligosaccharides through to insoluble, non-fermentable, cellulose-type fibres. Fermentation to SCFAs creates osmotic load and is accompanied by gas production [50]. In the colon, fibres increase colonic biomass by stimulating colonic bacterial proliferation and retaining fluid [50]. The solubility and fermentability of fibres influence the nature and extent of their bulking capacity, SCFA production and gas creation.

The mechanisms and metabolic consequences of fibre degradation in the colon have been well characterised across the spectrum of fibre types, but less is known about the extent and consequences of fermentation in the duodenum. Acetate and other SCFAs can be absorbed and metabolized in the proximal small intestine, and emerging evidence suggests that fermentation of SCFAs may occur in the upper gastrointestinal tract [51].

The influence of insoluble, non-fermentable fibres on symptom induction or dietary management of FD has not been reported. With a known gastrointestinal prokinetic effect through stimulation of digestive tract lining, it is feasible that increasing insoluble fibre may be beneficial in FD management through a similar mechanism to pharmacological prokinetics [52]. However, other food components that hasten gastrointestinal motility (caffeine in tea and coffee) have been reported to induce FD symptoms [21]. Wheat is a major source of insoluble dietary fibre in many populations and cultures, but the specific functionality of the insoluble fibre in wheat is yet to be differentiated from gluten or fructan FODMAPs in FGID management [49].

#### 4.2.1. Dietary Carbohydrate Modification Based on Diagnostic Tests

The gastrointestinal pain symptoms of sucrase-isomaltase deficiency overlap with FD symptomology. Congenital sucrase-isomaltase deficiency (CSID) is rare and is usually diagnosed in childhood, but there is increasing research interest in SI heterozygous individuals who present with the typical presentation symptoms of abdominal pain and bloating as well as watery diarrhoea. CSID can be diagnosed from hydrogen breath testing after an oral sucrose load or by disaccharidase assay of duodenal or jejunal mucosa obtained at endoscopy [53]. A 2015 study involving six paediatric patients reported minimal symptom improvement following dietary management advise, but a marked reduction in CSID symptoms with sacrosidase administration, with no adverse events, indicating that sacrosidase is an effective and well-tolerated treatment for patients with congenital SI deficiency [53].

It is conceivable that people with a low concentration of this enzyme may exhibit pain symptoms as a result of sucrose-induced osmotic imbalance [54]. This hypothesis has been investigated in IBS [54] and in SI heterozygotes, but it has not been explored in FD. Sucrose itself has not been identified as a common symptom-inducing food in FD, so is considered a less likely trigger in FD than other nutrients. However, it is useful for dietitians who manage FGIDs in primary care to be aware of potential SI deficiency, as an increasing proportion of people with FGIDs present having had breath testing for short chain carbohydrate absorption, or report having been ‘diagnosed’ with malabsorption or carbohydrate metabolizing enzyme deficiencies.

#### 4.2.2. Carbohydrates and Small Intestinal Bacterial Overgrowth

Small intestinal bacterial overgrowth (SIBO) may be found in patients with functional dyspepsia [55], although the relevance to symptoms is less clear. People who present to a dietitian who report that they have SIBO may have been diagnosed from culture of a duodenal aspirate and coliform counts or by a less accurate glucose or lactulose hydrogen breath test (false positives are common with breath testing) [56]. A bacterial culture count from duodenal aspirate is less common but considered the best diagnostic method, with a count of over 10^3^ (or in the older literature 10^5^) coliform units per milliliter considered abnormally high and indicative of SIBO [56].

People diagnosed with SIBO may have received antibiotic treatment to try and reduce the increased bacteria in the small intestine. Systemic antibiotics or the non-systemic antibiotic rifaximin can suppress SIBO [56]. SIBO antibiotic treatment may be accompanied by recommendations for dietary modification, most often involving a combination or variation on the low FODMAP diet or specific carbohydrate diet [47]. To date, no adequately powered placebo controlled trials have been conducted to determine whether dietary modification during or after SIBO antibiotic treatment confers any additional benefit, either in treatment efficacy or maintenance of symptom relief in FD or IBS [31,57].

People who consult with dietitians for FD or FD-like symptoms, who report having been diagnosed with, or suspected of having SIBO often have not had formal SIBO testing. They are more likely to have trialled a low FODMAP diet or a variation of the specific carbohydrate diet. These approaches may have controlled symptoms but not resolved the condition. In this situation, it is uncertain if SIBO testing is warranted.

#### 4.2.3. Carbohydrate and Fibre Considerations in Dietetic management of FD

A short term low FODMAP diet trial may be appropriate for engaged, motivated people with FD who report symptoms following ingestion of high FODMAP foods or with suspected SIBO. Two weeks of a low FODMAP diet should be adequate to determine responsiveness, if the client is moderately adherent to the low FODMAP diet [32,58].

Alternately, a limited exclusion diet is a suggested approach, whereby the most ‘likely suspect’ foods are removed and symptoms are monitored over two to four weeks. For either approach, it is recommended that the client records their food intake and symptoms, preferably using a food diary app that is accessible by the dietitian for monitoring and feedback (see Table 1 and Figure 1).

Specific manipulation of fibre types and amounts may be considered. The dietetic goal of increasing fibre may be symptom relief or dietary adequacy. Low FODMAP, high soluble fibre foods or a supplement could be trialled in conjunction with a low FODMAP diet if the client reports partial symptom relief with a low FODMAP diet. These foods or supplements could be included in the diet routinely if they do not induce symptoms, with the goal of maintaining adequate fibre intake with symptom minimization.

If a person with FD in not responsive to a low FODMAP diet, does not report adverse effects from eating wheat and has reported delayed gastric motility, short term manipulation of insoluble fibre (e.g., wheat bran) may be worth trialing. The outcome of this trial will be instructive to the dietitian, whether the client reports symptom reduction or exacerbation or no change to symptoms in response to wheat bran supplementation (see Table 1 and Figure 1).

### 4.3. Do Specific Food Proteins Induce FD Symptoms or Does Removal of Specific Proteins Alleviate Symptoms?

The growing body of evidence for immune involvement in FD suggests that food antigens may contribute to aetiology, in at least a subset of people with FD [59]. Increased mucosal eosinophils, intraepithelial cytotoxic T cells and systemic gut-homing T cells in the duodenum have been reported in people with FD [59], with a matching spectrum of mechanisms proposed for food protein involvement [60].

In a classic food allergy response, ingested food antigens (most often protein-derived) initiate mast cell and eosinophil recruitment and immunoglobulin E (IgE) production by B cells. Mast cells are activated by IgE resulting in their degranulation and the release of histamine and other inflammatory markers, producing allergy symptoms [59,61].

Elevated food antigen-specific IgG and challenge-induced eosinophil activation suggest atypical food sensitivities in a subset of people with FD [62]. Thirty percent of people diagnosed with FD report wheat or gluten sensitivity, making this the condition’s prominent food protein-related association [49]. Non coeliac gluten (or wheat) sensitivity (NCG/WS) is characterised by symptom induction following wheat ingestion, and symptom reduction with wheat exclusion. A subset of people with NCG/WS fit the coeliac HLA haplotype and serological antibody profile, subtle immunological changes and some pathological changes but lacking the villous atrophy that characterises coeliac disease [60].

Hypotheses for NCG/WS aetiology and symptom profile include subtle immune responses and ‘leaky’ tight junctions [60]. A recent review of immune mechanisms in FD reported consensus for increased T cells and ‘gut-homing’ T cells in FD, suggesting loss of mucosal homeostasis, possibly attributable to altered gastrointestinal microbiota [16]. Increased duodenal eosinophils and mast cells were a prominent finding, with a Th17 cytokine response being proposed as a possible explanatory factor in FD symptom induction [2]. Gluten derivatives are the primary candidates implicated in these subtle immune responses. Alternative wheat proteins amylase/trypsin inhibitors (ATIs) and wheat germ agglutinins have also been implicated. For example, Junker et al. (2021) identified two wheat ATIs as strong activators of innate immune responses in monocytes, macrophages, and dendritic cells. While not specific to FD, this finding suggests that ATIs may fuel inflammation and immune reactions in intestinal immune disorders [63].

The ‘leaky’ tight junction hypothesis involves gliadin as a Toll like receptor (TLR) ligand that may increase zonulin secretion, resulting in increased epithelial permeability and decreased barrier function [59]. There is substantial debate in the gastrointestinal research community about serum zonulin as a biomarker of intestinal permeability, with some suggesting that intestinal permeability is better assessed using dual-sugar assays or with immunohistochemistry and expression profiles of zonula occludens proteins [64].

An alternative hypothesis for dietary protein involves L-Glutamine, which can be depleted in infection or illness [65]. Clinical and experimental studies have demonstrated that glutamine supplementation reduces intestinal permeability from various stressors and helps maintain the normal intestinal barrier function in GI conditions [65,66]. Although not yet applied to FD, oral dietary glutamine powder supplementation (5 g/three time daily) was reported to reduce intestinal permeability, and reduced IBS-related endpoints in IBS-diarrhoea in one study [67].

It is important that dietitians working in the FGID space are aware of the ‘leaky gut hypothesis’. In the authors’ experiences, a high proportion of people with FD and other FGIDs who are seeking dietary management may have self-diagnosed or been advised by complementary therapists as having a ‘leaky gut’. These clients may have trialled ‘gut building’ remedies, and therefore the dietitian needs to understand the basis for such approaches, be capable in differentiating evidence-based approaches, and skilled in communicating about this with clients as part of the overall dietary management plan.

Other than wheat proteins, other dietary protein modifications that have been investigated in relation to FD symptom management are cow’s milk protein and a low protein diet. There is very limited literature to inform a dietary management approach for people who report symptoms after ingestion of cow’s milk in cases where cow’s milk-specific IgE or lactose intolerance have been investigated and excluded [60]. A protein-modified diet has been proposed as a means of altering colonic microbiome towards an anti-inflammatory profile, and a low intake of aromatic and sulphur-containing amino acids is hypothesised to modulate hydrogen sulphide production, with possible implications in reducing visceral hypersensitivity [47]. These hypotheses are yet to be investigated in suitably powered intervention studies that also account for the influence of fatty acids such as capric acid.

The symptoms of FD and NCG/WS overlap with characteristics that some alternative therapists diagnose as ‘adrenal fatigue’. These non-specific symptoms include cortisol imbalance, tiredness, sleep disturbances, salt and sugar cravings. There is no substantiation that adrenal fatigue is a true medical condition [68], but dietitians managing FD in primary care are better positioned to manage FD if they are aware of such alternative treatment approaches.

While removing wheat from the diet may seem to be the natural and reasonable approach for those reporting wheat sensitivity, determining whether the sensitivity relates to a wheat protein or carbohydrate (FODMAP/fibre) is important so that diet is not restricted more than necessary. Careful dietary elimination and food challenges can differentiate gluten and FODMAPs so the food range can be broadened as far as possible while maintaining adequate symptom relief [69]. Dietitians facilitating FD management need a working understanding of immune responses to food antigens to distinguish which (if any) dietary protein modifications are appropriate for people with FD, and how to implement dietary modifications while maintaining nutritional adequacy.

### 4.4. Do Specific Dietary Fats or Total Intake of Fats Induce or Alleviate FD Symptoms

In addition to their role in increasing satiety and slowing gastric motility (described in Section 3.2), dietary fats may influence FD pathogenesis or symptom induction directly or indirectly via microbiota, enzyme or bile acid signaling, or a combination (or overlap) of these mechanisms.

As microbiota substrate, total dietary fat intake and dietary fat profile influence the relative abundance and diversity of duodenal bacteria, with high fat intake associated with increased Firmicutes and reduced Bacteroidetes phyla [70]. In FD specifically, high *Streptococcus* and low *Prevotella, Veillonella* and *Actinomyces* have been reported, with *Prevotella* restoration associated with improved PDS symptoms [71].

The links between bile acid biology and FD pathophysiology may implicate duodenal microbiota-related bile acid signalling in FD. Lower duodenal mucosal resistance in FD compared to healthy controls is correlated to duodenal bile acid pool composition [72]. Keely et al. (2020) hypothesise that bile acid alterations potentially link diet, the microbiota and mucosal barrier dysfunction, all of which are likely important determinants of FD. In support of this hypothesis, CCK is increased in FD after a high fat meal and is known to stimulate bile acids and digestive enzymes [73].

Consumption of excess dietary fats may differentially increase intestinal permeability by modulating expression and distribution of tight junctions, stimulating a shift to barrier-disrupting hydrophobic bile acids, and inducing epithelial cell oxidative stress and apoptosis. A high-fat diet also enhances intestinal permeability directly by stimulating proinflammatory signalling cascades and indirectly via increasing barrier-disrupting cytokines, decreasing barrier-forming cytokines, negatively modulating the intestinal mucus composition and enriching the gut microflora with barrier-disrupting species [74]. Conversely, the anti-inflammatory effect of a Mediterranean diet has been linked with beneficial changes in the stool microbiota composition, including increases in *Bacteroides* and *Clostridium* genera and decreases in Proteobacteria and Bacillaceae [75].

While some of these factors relate to pathophysiology and others relate more directly to symptom induction or relief, all are important dietary management considerations in FD. Other considerations are the influence of fat-binding medications and soluble fibres, which may be relevant to microbiota substrate availability, although this is more likely to affect the lower gastrointestinal tract than the small intestine. Based on the available evidence, a low saturated fat, Mediterranean-style diet seems preferable to maintain upper gastrointestinal mucosal integrity, a desirable microbiota profile and optimal metabolic signaling. Dietetic management of FD may involve a short-term trial of reduced dietary fats, particularly saturated fats. The differential dietary management pathway specific to fats is shown in Figure 1.

### 4.5. Anti-inflammatory Approach to FD Dietary Management

Subtle immune activation and inflammatory responses are a key component of the duodenal microbiome hypothesis in FD pathogenesis [16,59,76]. Intestinal eosinophilia and mastocytosis may be exacerbated by diet-induced microbiota community profiles, but a proinflammatory diet is more likely to be a trigger than an underlying cause of FD [59]. The potential for an ‘anti-inflammatory’ diet to ameliorate the low grade inflammatory state in FD has not been studied. In other gastrointestinal conditions, consumption of a pro-inflammatory diet was associated with increased odds of IBS in one study [77], and switching from a ‘proinflammatory’ to ‘anti-inflammatory’ diet for six months prevented colonic inflammation compared to healthy eating advice in inflammatory bowel disease in another study [78]. By extension of this hypothesis, there may be a potential role for a similar approach in FD management. For the majority of people with FD, a predominantly ‘anti-inflammatory’ diet (high in fibre, prebiotics, probiotics, antioxidants and omega-3 fatty acids and low in saturated fats, red meat, sugar and alcohol) is consistent with the other dietary strategies proposed for FD in this paper, with some modification based on primary symptomatology.

## 5. Micronutrients and Additives in FD

### 5.1. Natural Food Chemicals

Naturally occurring food chemicals are endogenously produced by plants and animals for functional roles such as preservation and are consumed by humans in food. In sensitive individuals, natural food chemicals can induce non-allergic hypersensitivity by stimulating nociceptors. Each person with food sensitivities reacts differently to natural food chemicals, but the most common symptoms are altered bowel habits, abdominal pain, bloating, headaches, migraines, fatigue, behavioural problems or urticaria [48,79].

The natural food chemicals of most relevance to dietary management of FGIDs are salicylates, amines, glutamates and lectins. Salicylates, which are predominantly from plant foods have been implicated in IBS symptom induction [80]. Biogenic or vasoactive amines are produced by bacteria during fermentation, storage or decay. Among the many types of amines, histamine is the most often linked to food-related symptoms [81]. It is also hypothesised that some foods cause histamine release directly from tissue mast cells [82]. However, histamine levels in foods differ with ageing, fermentation and decay, making it difficult to assess associations between dietary histamine (or histidine) and gastrointestinal symptoms [81].

Glutamates are consumed either as naturally occurring food chemicals or in food additives, particularly monosodium glutamate. Some individuals experience gastro or extra-intestinal symptoms from glutamate ingestion, although usually only after eating amounts greater than usual dietary intake [81,83]. Glutamates have not been examined in relation to FD, but have the potential to trigger FGID symptoms via chemoreceptors or mast cells [84].

Lectins are carbohydrate-binding proteins that perform important biological roles and are common in food plants but are somewhat digestion-resistant [85]. Based on the hypothesis that lectins can bind gut bacteria and epithelial cells, releasing endotoxins that induce intestinal permeability and allow lectins to penetrate the digestive barrier and bind to body tissue inducing inflammatory and autoimmune responses [85]. There is scant evidence for dietary lectin involvement in FD aetiology or symptom induction, but dietitians need to be familiar with the concept due to the popularity of dietary lectin restriction as a management approach to auto-immune and ‘leaky gut’ conditions despite a lack of evidence of efficacy.

Natural food chemicals can induce some gastrointestinal symptoms that overlap with IBS, but no controlled trials have assessed low chemical diets in FD symptom induction or management [47]. The overlap of food triggers and aetiopathology between true food allergies and food chemical sensitivities is challenging for dietitians to navigate in FGID management. An important aspect of the dietetic management role is to weigh up the low level of evidence for a low food chemical diets in FD management and risk of nutritional deficiencies associated with elimination diets, against the presenting symptoms, diet history and client motivation to determine whether a low food chemical trial is appropriate. Co-presenting extra-intestinal symptoms that are consistent with natural food chemicals, such as migraine would be the most likely reason to trial a low food chemical diet. Examples include history of migraine headaches associated with chocolate, cheeses or aged meats, or adverse reaction to aspirin medication. Close monitoring of symptoms throughout the trial and dietary challenges would help to determine any gastrointestinal association, with the aim of achieving a threshold balance between ingestion and symptom minimisation.

### 5.2. Food Additives in FD Aetiology and Symptom Induction

The term food additives encompasses any natural or artificial item that is added to a food or recipe to improve the taste or appearance, keeping quality, stability or preservation of a food [86]. Food additives that are relevant to FD dietary management include glutamates, microbial transglutaminase (mTG) and some food colourings and flavouring agents. The potential role of food additives in FGID manifestation is highlighted by recent studies focused on ultra-processed foods, which are food- and additive-derived formulations developed to be convenient and hyper-palatable, and often displace unprocessed, fresh food [87]. Schnabel et al. (2018) reported an association between overlapping FD/IBS (Rome III criteria) and the proportion of diet classified as ultra-processed foods from three 24 h recalls in a sample of 33,343 participants in the prospective observational NutriNet-Sante study [88].

At a public health level, the European Food Standards Authority has set standards for the use of glutamates used as food additives based on the highest dose at which scientists observed no adverse effects on test animals in toxicity studies [89]. This standard has not been applied universally, but does show that public health efforts to decrease risk and symptoms from consumption of excess food additives. In parallel, the European Food Standards Authority’s 2015 emerging risk report identified the need for urgent research into the long terms health effects of food emulsifiers on gut microbiome and inflammation [90].

Microbial transglutaminase (mTG) is particularly interesting because food standards authorities classify mTG as an industrial processing aid rather than a food additive, so it is not regulated in the same manner. Transglutaminases (TGs) play a crucial role in physiological homeostasis, exerting their biological functions by deamidation or cross-linking of proteins. Microbial transglutaminase (mTG) is secreted by a range of microbes, of which *Streptoverticillium mobaraense* is the most frequently used in the food industry. The food manufacturing benefits of mTG enzymatic action include improved emulsification, consistency, texture, shelf life and palatability [91]. Food producers consider mTGs to be safe, non-toxic, non-allergenic, non-immunogenic and non-pathogenic for public health [6]. However, emerging epidemiological, scientific and clinical mTG research indicates potentially detrimental effects based on mTGs protein modifying abilities, which functionally imitate endogenous tissue TG [91]. Although no studies have specifically investigated TGs in functional dyspepsia, we flag this as a research need based on FGID manifestation in response to food additives [87] in conjunction with the role of TGs in inflammatory intestinal diseases [92].

Artificial sweeteners are sugar substitutes with low or no energy (calorie) content, and include acesulfame, aspartame, neotame, saccharin, and sucralose. There are virtually no data on alterations in permeability, immune activation, and visceral sensation in FGID. Animal studies suggest that artificial sweeteners affect the gut microbiome, but the human studies conducted have been too heterogeneous in terms of study populations and doses to be synthesized and compared with animal study findings [93]. Increased incretin secretion in response to artificial sweetener consumption is a proposed mechanism for changes in GI motility, but this has only been reported to be increased in one human study where sucralose was combined with glucose. In their 2017 review, Bryant and Mclaughlin reported that human gut exposure to artificial sweeteners does not replicate any of the effects on gastric motility, gut hormones or appetitive responses evoked by caloric sugars.

Other non-nutritive sweeteners, such as sorbitol, xylitol and stevia, are not technically considered to fall under the term ‘artificial sweeteners’ [93]. A high proportion of these sweeteners are polyol FODMAPs and therefore can impact on GI symptoms in FGIDs [94]. The mechanism and dietary management implications of polyol consumption is discussed in Section 3. Many foods contain both artificial and non-nutritive sweeteners, making it complicated to conduct human studies to differentiate which sweeteners could be a potential cause of GI symptoms.

As with natural food chemicals, there may be some benefit in the removal of a specific food additive from the diet for a limited period for individuals who report specific, consistent adverse reactions to foods with a known additive [81]. Any dietary intervention, whether for the purposes of diagnosis or management of food allergy or food intolerance, should be adapted to the individual’s dietary habits and a suitably trained dietitian should ensure nutritional needs are met. Ultimately a healthy diet should be the aim for all patients presenting in the clinic [70].

## 6. Dietary Influences on Microbiota in FD

Microbiota profiling in FD has been limited to stool sampling or contamination-prone duodenal mucosa collection methods [76,95]. Recent advances in sampling methods have resulted in aseptic collection of duodenal mucosal associated microbiota, with the primary genera reported (in descending abundance) as *Streptococcus, Prevotella, Porphyromonas, Veillonella, Fusobacterium, Neisseria, Granulicatella, Leptotrichia* and *Haemophilus* [95]. Inconsistencies in study findings about dietary influences on microbiota in FD reported may relate to differences in sampling and collection methods.

Studies investigating diet-related differences in gastrointestinal microbiota composition have generally indicated that people who consume a diet that is high in animal protein and saturated fats and low in carbohydrate and fibre have a microbiome profile that is higher in *Bacteroides* and lower in *Prevotella* than those with the opposite dietary macronutrient balance [47]. In a 10-day feeding study, Wu et al. reported that enterotypes were strongly associated with protein and animal fat (*Bacteroides*) versus carbohydrates (*Prevotella*), that microbiome composition changes were detectable within 24 h of commencing a high-fat/low-fiber or low-fat/high-fiber diet, but that enterotype identity remained stable throughout the study [96]. In a cross sectional study that analysed relationships between gut microbiome and host factors in 3400 healthy individuals in the United States (US), Moran et al. (2020) also observed the dominant *Firmicutes-Bacteroides* axis and dichotomy between *Bacteroides* and *Prevotella,* and suggested that the inverse relationships between putatively beneficial bacteria may relate to competition for nutrient substrate [97]. Interestingly, an abundance of one *Erysipelotrichaceae* genus was significantly correlated with self-reported symptoms of bloating and nausea in the healthy study population [97].

The small bowel is more acidic, has higher oxygen levels and higher antimicrobial peptides than the colon. Along with phasic propulsion at the ileum, these factors limit bacterial densities in the proximal small intestine. The upper gastrointestinal tract is dominated by fast-growing facultative anaerobes like *Lactobacillaceae* and *Enterobacteriaceae* families that tolerate this environment and can metabolise the available simple carbohydrates [98].

Bacterial abundance and prevalence in three FD-specific studies have reported microbiota profiles that are consistent with expected small intestinal profiles, and also consistent with a dietary profile that is higher in dietary fats and protein and lower carbohydrates and fibre.

Nakae et al. (2016) reported a lower *Prevotella* abundance and higher *Bifidobacterium* and *Clostridium* abundance in an FD cohort compared to gastric fluid microbiome of healthy individuals [99]. Zhong et al. (2017) reported increased relative abundance of *Streptococcus* and reduced anaerobic genera *Prevotella*, *Veillonella* and *Actinomyces* in FD patients compared to matched healthy controls [76] and Fukui et al. (2020) reported increased *Streptococcus* in the upper gut in the FD cohort [100]. Microbiota changes correlated with upper GI symptoms in two of these studies [76,100]. These findings indicate that microbiota may influence FD aetiology and symptomatology, and these are parallels between FD symptom-microbiome and diet-microbiome interactions.

Findings that further implicate the microbiome in FD include associations between bacterial profiles and load in the small intestine and FD symptom induction [4], previous antibiotic treatment being a risk factor for developing FD [101,102] and the non-absorbable antibiotic rifaximin in one randomized placebo controlled trial was superior to placebo for treating the condition [103].

Direct alterations to bacterial composition or load are potential mechanisms that may induce FD symptoms. Increased duodenal mucosal bacterial load and decreased diversity were correlated with increased meal-related symptoms during a nutrient challenge test in one study [76], and probiotic supplementation restored *Prevotella* abundance and reduced PDS symptoms in people with FD in another study [99]. In a mouse study, in vivo β2→1-fructan modulation of the immune system was partly dependent on microbiome, with different immune response types between short- and long-chain β2→1-fructans [104]. These results indicate that altered macronutrient metabolism may be associated with FD symptom induction [59], and provide further credence to macronutrient manipulation as an FD dietary management strategy. Bile acid-mediated mechanisms provide an alternative explanation and are described in Section 5.

Probiotic supplementation may have a therapeutic role in FD via microbiome modulation. Three studies have reported improvement in FD symptoms after probiotic *Lactobacillus gasseri OLL2716* supplementation [71,105,106] and one study reported a post-probiotic shift in FD patients towards microbiota composition seen in the healthy volunteers, with an increased abundance of Proteobacteria than Bacteroidetes, and presence of Acidobacteria [71]. Prebiotic and probiotic supplementation have also been shown to ameliorate the reduced bacterial abundance resulting from a low FODMAP diet used to manage IBS symptoms [47], indicating that microbiome is implicated across the spectrum of FGIDs.

A novel consideration for future FD-microbiota research relates to postbiotics (or metabiotics). Postbiotics are substances released by or produced through the metabolic activity of microbiota which exerts a beneficial effect on the host, directly or indirectly with effects on the human health and include SCFAs, enzymes, peptides, teichoic acids, vitamins and plasmalogens [107]. Some of the mechanisms of FD symptom induction or alleviation described in this section relate to the metabolites from diet-microbiota processes. While postbiotic dietary supplementation have not yet been described in FD management, the concept of dietary supplementation with metabolites of known benefit in FD symptom control warrants exploration.

Medications that target nutrient absorption and metabolism can influence microbiota abundance and diversity. Manor et al. (2020) reported that blood glucose medications and cholesterol-lowering medication use is associated with altered microbiota abundance, and that blood glucose lowering medications were associated with an enriched metabolic pathway for fructose metabolism [97]. As fructose is FODMAP that influences intestinal osmotic balance, this has potential implications for FD symptoms.

Together, these findings indicate solid, hypothesis-driven foundations and emerging evidence for mechanistic relationships between microbiota community profiles or specific bacterial genus influence FD aetiology or symptoms. These FD-microbiome mechanisms triangulate with the influences of dietary patterns on aetiology-related bacterial profiles, and of specific nutrients on FD symptom induction or relief. There are exciting but very preliminary dietary management implications for this duodenal bacteria-FD-diet triangulation that are likely to progress towards personalised dietary management plans that are individually tailored to reduce symptoms by altering microbial profiles. Access to affordable bacterial community profiling will be a limiting factor in this field in the medium term, although an increasing proportion of people with FGIDs are completing bacterial community profiling through commercial services [108,109].

## 7. Complementary Therapies and Micronutrient Supplementation in Dietary Management of FD

Despite extensive investigation into the dietary and nutrient intakes of people with FD, there is currently no evidence of underlying micronutrient deficiency causation [110]. This may reflect the absence of a relationship between specific micronutrients and FD pathophysiology or symptomatology, or could be attributable to the predominantly cross sectional nature of dietary assessment in FD to date, the variability in dietary management approaches or underpowering of studies to detect differences in micronutrient intake [21]. Dietitians need to consider overall micronutrient status and adequacy as part of their dietary management approach for clients with FD, rather than expecting micronutrients to play a pivotal role in symptom induction or management.

Complementary and alternative treatments have been reported as beneficial in treating FD symptoms [27], and may thereby enhance the effectiveness of dietary management strategies. Four trials have reported that supplements containing peppermint and caraway oil were more effective than placebo in improving dyspeptic symptoms, with an average decreased intensity of epigastric pain compared to placebo [4,111,112,113]. In mechanistic studies, the active components of these oils were reported to have antiemetic, choleretic and spasmolytic effects in the distal stomach and duodenal bulb [112]. The anti-inflammatory, antiemetic and motility-related properties of polyphenols in ginger enhance gastric emptying, improve gastric motility, reduce nausea and vomiting and reduce inflammation [112]. Iberogast (STW5) contains a combination of nine herbs and appears to reduce FD symptoms by enhancing antral motility and proximal gastric relaxation [38].

Conversely, some herbal treatments prescribed for gastrointestinal symptoms are sometimes high in FODMAPs, and therefore need to be considered with caution or discontinued during FD dietary trials and challenges. Herbal teas that are high FODMAP include chrysanthemum, chamomile, dandelion, fennel, oolong, some fruit teas and chicory root [114].

Overall, some herbal therapies are considered to offer significant benefits to people with FGIDs [115]. As complementary treatments can be adjunctive to dietary management, it is important for dietitians to understand the respective mechanism of action for each herbal therapy so they can tailor the overall diet and herbal therapy approach to the symptoms of each individual client.

## 8. FD that is Unresponsive to Dietary Management

Gastrointestinal symptoms sometimes persist despite dietary and medical intervention. Best-practice dietetic FGID management includes ongoing recognition and reference to non-diet management approaches that could be considered as complementary or subsequent to dietary intervention. Treatment that focuses on gut-brain axis are a logical companion to dietary measures. Clients may be willing to consider a psychological approach if recommended by their treating dietitian, with whom they have established a therapeutic relationship. Gut-brain interactions are bidirectional, so perceived stress and corticotropin-releasing hormone pathways that influence gastrointestinal function, including permeability [116], may be particularly relevant for people with FD who report co-existing anxiety. Gut-focused hypnotherapy is reported to be highly effective in the long-term management of FD, with a dramatic reduction in medication use and consultation rates after 56 weeks in a study involving 126 FD patients randomized to hypnotherapy, supportive therapy plus placebo medication, or medical treatment for 16 weeks [117]. In another study of 100 FD patients with refractory symptoms, intensive medical management that incorporated a psychological intervention resulted in superior long-term-outcome than standard care [118]. Exposure and mindfulness therapy [119], acceptance and commitment therapy [120], and cognitive behavioural therapy [121] are other psychological treatment approaches that have been successfully tested in IBS but not yet applied to FD. Therefore, optimal dietetic management of FD requires an integrated view of FD as a disorder of brain–gut signalling [122], with psychological factors (such as personality traits and anticipatory anxiety) considered when assessing symptom responses to food and eating.

## 9. Guidance on FD Dietetic Management based on Existing Research

Having first been characterised in the early 1990s, FD is a relatively ‘new’ GI disorder. Its diagnostic criteria have been refined three times since the original Rome Criteria were set, and now acknowledge that this ‘functional’ upper GI disorder is extremely multifaceted, and likely to have immunological, neurological and inflammatory components rather than being purely functional or strictly gastrointestinal. FD is specific to the upper small intestine, a difficult part of the digestive tract to investigate. Combining these factors with the complexity of small intestinal functions (and malfunctioning), it is understandable that there remains a lot to learn about FD.

To date, dietary interventions and studies of dietary management in FD have been limited by methodological problems that relate to the challenges of using self-reported tools for the assessment of both dietary intake and FD symptoms [123]. Our research team has highlighted the need for more objective measures of dietary intake and FGID symptom assessment, and better alignment between the data collection methods for diet and symptoms [123].

In a comprehensive review of dietary therapies for functional gastrointestinal disorder symptoms, Tuck and Vanner (2017) provide a succinct summary of the evidence for specific biomarkers to predict responses to specific dietary therapies for the spectrum of FGID pathophysiological indicators [47]. They describe objective genetic, immune and microbiota biomarkers of FGIDs that are highly applicable to the specific condition of FD. Concomitant advances in image-based, nutrient database-linked dietary intake reporting and analysis platforms have transformed dietary assessment [124].

It is conceivable that understanding of diet in FD will evolve quickly in response to these recent rapid advancements in technologies for monitoring and assessing dietary intake, for identifying and quantifying duodenal mucosal-associated microbiota and related to FD biomarkers. These parallel changes will facilitate more robust assessment of dietary intake in relation to FD aetiology, symptom induction and dietary management.

In this review, we have collated the available information about diet-related aspects of the aetiology, pathophysiology and symptomology of FD. The dietary factors related to FD have been presented as a dietary management matrix. There are no randomised controlled trials to guide clinical practice in terms of dietary management in FD, and such trials should be a future priority. In this section of the paper, we provide expert guidance around identifying, selecting, prioritising and administering FD dietary management strategies and plans. We also provide recommendations to address predisposing dietary patterns implicated in FD and role delineation and collaboration between the GP, GE and dietitian in FD dietary management.

### 9.1. Establishing Therapeutic Dietary Management Relationship

Eighty percent of FD sufferers, which equates to approximately 8% of the adult population, experience eating-related symptoms [18]. With food being such a dominant factor in FD, it is unsurprising that individuals referred to GEs for treatment expect to discuss dietary management during consultations. The disconnect between patient and GE expectations and actions in relation to dietary management reported by Farrall et al. (2009) may be linked to time limitations and prioritisation or perceived scope of practice (i.e., the gastroenterologist does not consider dietary management as their role) [33]. It may also reflect the need for a clearer and better articulated referral pathway, whereby GEs are upskilled and well-resourced to provide preliminary or priming dietary advice, and to advocate for and refer patients to specialist FGID dietitians. In turn, dietitians need to demonstrate and promote their capability in meeting the needs of clients and expectations of referring GEs in FGID management. The ‘dietitian-first’ model for IBS management provides a template that could be applied to FD, but would require a stronger and clearer evidence-base and equivalently skilled dietetic workforce [26].

For all dietary management strategies, communication of dietary intake information and advice between the client and dietitian is crucial, and relies on a trusting, empathetic and collaborative therapeutic relationship, in addition to dietetic skill and knowledge. It is ideal to discuss and agree on how dietary intake information will be communicated in the initial consultation. The dietitian might show the client an app they can use (some of which are linked to a platform accessible in real-time by the dietitian), provide a food diary to complete or provide a meal plan that the client follows and reports on adherence to. These tools increase the effectiveness of dietetic consultation time, as the dietitian has access to accurate information and the client has a reminder of eating occasions. Showing the client how these tools will be used and using the information the client collects as part of the consultation, is likely to increase the client’s compliance with dietary intake data collection.

### 9.2. Carbohydrate, FODMAP and Fibre-Focused Dietary Management of Functional Dyspepsia

For a client who presents with uncomplicated PDS-subtype FD, a two-week, dietitian administered low FODMAP diet trial may be warranted. If the client’s symptoms reduce substantially, sequential challenges will determine whether specific FODMAPs induce symptoms or whether overall FODMAP load induces symptoms. If specific FODMAPs are implicated, it is helpful to assess whether these are predominantly the ‘osmotic-type’ polyols, fructose and lactose or the ‘gas-producing’ oligosaccharides, as this may inform the management approach. For example, if oligosaccharides are problematic, this implicates the duodenal microbiota. Symptom induction after eating fluid-drawing FODMAPs implicates visceral hypersensitivity. Differentiating between these is helpful for subsequent diet advice and management, but is not always definitive. If the overall FODMAP load is implicated, further dietary adjustments can be trialled to determine the individuals FODMAP threshold. Supplementary education about increasing the ‘FODMAP threshold’ through diet, microbiota and lifestyle management is recommended as part of the dietary management process.

If a low FODMAP diet trial is not effective for reducing FD symptoms management, the dietitian would assess compliance to determine whether an extension of the trial or another approach is warranted. If compliance is difficult but the low FODMAP diet trial shows promise for symptom reduction, a modified or ‘progressive’ FODMAP reduction may be more suitable and can be negotiated between dietitian and client. It is important for the dietitian to be clear that if a client’s symptoms are unchanged or only slightly reduced after the FODMAP trial, that is instructive for the next management approach, helping to focus the dietary management plan. A subset of clients are not fully responsive to a low FODMAP diet but may benefit from a combined approach that also involves a stress management component (e.g., mindfulness, yoga, meditation or progressive relaxation for anxiety management).

The efficacy of a low FODMAP diet as an antibiotic alternative or post antibiotic treatment for SIBO is unclear. For clients who present reporting some form of FODMAP restriction, the dietitian is well-positioned to ensure that nutritional adequacy is maintained throughout antibiotic treatment or until adequate symptom relief is achieved.

Parallel or subsequent to dietary carbohydrate manipulation for FD symptom management, dietary fibre intake is a key FD management priority. High dietary fibre intake is associated with prebiotic and ‘postbiotic’ gastrointestinal benefits and broader health status. A high-fibre, low FODMAP diet is possible with selective choices and portions of grains, fruit and vegetables. These food-based choices may be supplemented with high fibre foods such as chia seeds or low FODMAP, high soluble fibre supplements (often based on guar gum). Finding an optimal balance and amount of fibre while avoiding symptom induction is challenging but important for improving longer term management, especially for individuals with overlapping FD and IBS.

### 9.3. Protein, ‘Leaky Gut’ and Immune System Approaches to FD Dietary Management

The subtle inflammatory and other immune system responses characteristics of FD are not as simple to identify or trial dietary approaches on. Elevated eosinophils and mast cell immune responses are identified from tissue samples not blood tests, so it is not possible to identify from routine blood tests. Indicators that are available from the client’s medical history or previous testing could include: HLA haplotype and coeliac disease panel (blood test), history of atopy, family history of coeliac disease or cow’s milk allergy, any other food allergy or histamine response.

As gluten (or gliadin) sensitivity in FD is not the same as coeliac disease, any dietary trial to test wheat or gluten sensitivity would not need to be completely gluten free. The dietitian can support the client to first differentiate between gluten and FODMAP sensitivity by trialing a low FODMAP diet and adding a gluten challenge in low FODMAP challenge protocol [69]. Similarly, a suspected cow’s milk protein response can be differentiated from lactose intolerance by amending a low FODMAP protocol to include lactose-free milk and A2-protein cow’s milk challenges in addition to the cow’s milk challenge.

As increased intestinal permeability is associated with a worsening of symptoms and an increase in inflammatory immune responses to luminal proteins in GI disorders, a management pathway that supports the intestinal barrier should be considered. The possible benefit of L-glutamine supplementation may be warranted post-infection or acute illness, although the efficacy in FD remains untested. A high-fibre, low fat, modest animal protein diet to optimize the microbial enterotype is therefore longer term dietary strategy to protect the immune system.

A substantial proportion of clients who present for dietetic management of FD have already been prescribed or self-administered diets for ‘leaky gut’, ‘adrenal fatigue’ or ‘inflammation’, likely to include strategies of varying and sometimes questionable evidence. The FGID dietitian needs to maintain engagement with client and gradually educate towards an evidence based approach, bearing in mind that symptom relief is the key outcome, so any modifications that have a perceived benefit may be worth maintaining, as long as they are not detrimental.

### 9.4. Dietary Fats and ‘Anti-Inflammatory’ Diet in FD Management

A low saturated fat, high omega-3 diet Mediterranean-style diet seems desirable for reducing risk of FD and reducing total dietary fat may avert FD symptoms. A short term trial of reduced dietary fats, particularly saturated fats, is warranted where motility or bile acids are implicated in the presentation.

A diet that contains a modest amount of protein and alcohol, is high in dietary fibre, antioxidants, pre- and probiotic foods equates to an anti-inflammatory diet. In FGIDs, this dietary profile is recommended for optimizing the gut bacterial community profile but the diet needs to be individually tailored to achieve this nutrient balance while accounting for each person’s symptom-inducing foods and dietary preferences. The findings of Manor et al. (2020) that cluster-based predict individual-level responses to dietary changes [97] are also likely applicable to FGID population, but any changes in food or nutrients need to be achieved without symptom-induction. Achieving an ‘anti-inflammatory’ or ‘gut bacteria-friendly’ dietary profile while managing a FGID is a highly technical and mutable process that requires FGID skilled dietetic guidance.

The principles of an ‘anti-inflammatory’ or ‘Mediterranean-style’ diet are the basis of most dietary guidelines for general health maintenance and chronic disease prevention. Following these principles results in higher intake of ‘whole foods’ and lower intake of food additives, which is ideal regardless of the extent to which these are directly or indirectly implicated in FD.

Considering food chemicals and additives in the FD dietary management approach

Based on current evidence, an elimination diet to remove natural food chemicals or food additives would only be recommended as an initial FD dietary management approach if the client reports specific, consistent adverse gastro- and extra-intestinal reactions to foods high in known chemicals or additives. If both FODMAPs and food chemicals are implicated, the dietitian and client would need to discuss options for dietary approaches and prioritise which would be trialled first, and how to ensure that food preferences are catered for and to achieve nutritional adequacy.

In the general public and amongst some primary care clinicians, there may be confusion about which non-nutritive sweeteners are high in FODMAPs. For example, it is common for clients to present to a dietitian having been incorrectly advised that diet soft drink contains polyol FODMAPs. Polyols (mainly sorbitol, but also lactitol, mannitol and maltitol) are common non-nutritive sweeteners in sugar free chewing gum, confectionery, bakery, some dairy products and pastry items, but soft drinks and sports drinks tend to contain non-FODMAP sweeteners such as acesulfame K, sucralose, aspartame or cyclamate [125].

### 9.5. Delineation of Roles in FD Dietary Education and Advice

The GP or GE have crucial medical diagnostic, medical testing and referral roles in FD management. Although GPs can positively diagnose FD based on symptoms, red flag exclusion and relevant exclusionary tests, most clinicians regard FD as a diagnosis of exclusion, which leads to unnecessary early GE referrals and upper endoscopic investigations [126]. As tertiary GE clinics require referrals, and referrals are triaged based on strict criteria, people with FGIDs may not be consulted for years, creating a ‘bottleneck’ in FGID management [126]. The dietary management approach is much clearer if a client presents with a diagnosis of FD rather than reporting symptoms of FD to the dietitian incidentally during a consultation for another condition. Therefore, improved education for primary healthcare providers is pivotal in improving diagnostic processes in FD and other FGIDs, and could also be used to provide training in preliminary dietary assessment and dietetic referral as part of a shared care approach to FD management.

### 9.6. Training in FGID Dietary Management

FD dietary management involves matching dietary recommendations to specific symptom phenotypes, while also accounting for individual food preferences [47]. Personally tailored approaches are needed that account for the multi-factorial nature of FD, and accounts for all available aetiopathological information available to the clinician, instead of generic, condition-defined dietary advice. With increased numbers of individuals completing gut microbiome sequencing and nutrigenomic profiling, FGID-specialist dietitians would therefore be expected to be aware of the specific macronutrient-bacterial associations that are relevant to FD, be capable of interpreting individual microbiota analytic results and integrating the findings into personally tailored, microbiota targeted, dietary management plans for individuals with FD.

Tuck et al. (2018) predict that targeted dietetic approaches will potentially reduce the level of dietary restriction needed to optimize symptom management, with only reduction of ‘likely triggers’ [120]. To achieve this outcome, treating dietitians will have the ability to match dietary approaches to symptomatology and to prioritise associated management strategies. For this to be possible, dietitians needs develop a collaborative, trusting, sustainable and educational therapeutic relationship with their clients to resolve presenting symptoms, and so the dietary approach can be adapted if symptoms or client’s needs or circumstances change.

## 10. Conclusions

The complex biopsychosocial nature of FD demands that treating dietitians possess well developed education skills to tailor evidence-based information to individual needs and the tact to account for or accommodate any previous management approaches. This paper provides guidance on how progress towards a more differential dietetic management approach may be achieved in FD management, and recommendations on how clinicians involved in FD management can collaborate on improving models of care for FD. Improving FD dietary management will require ongoing progress in laboratory studies investigating FD microbiome, immunology and physiology and a much stronger evidence base from well-designed dietary interventions in randomized controlled trials. The findings from FD research will need to be applied at a primary care level, using similar models of care and training and education programs that have been successfully developed for IBS management. Quantum gains in understanding of the multi-factorial condition of functional dyspepsia have been achieved in the thirty years since it was first characterised. In this paper, we propose a dietary management oriented treatment approach to further progress and refine FD treatment.

## Figures and Tables

**Figure 1 nutrients-13-01109-f001:**
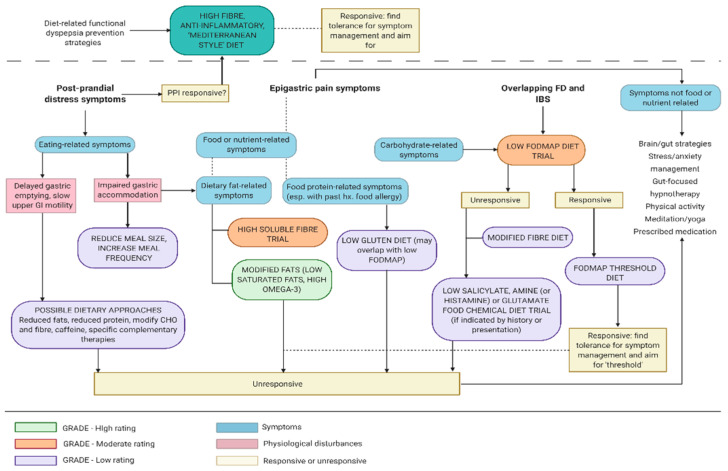
Functional dyspepsia dietary management flow chart based on food-nutrient-symptom presentations and informed by the Grading of Recommendations, Assessment, Development and Evaluations (GRADE) framework for clinical practice.

**Table 1 nutrients-13-01109-t001:** Functional dyspepsia dietary management matrix for stratification and prioritisation of dietetic strategy alternatives.

**Dietary Strategy** **---------------------------** **Considerations and** **Predisposing** **Or Risk Factors**	Regular Small Meals	Modified Texture	Reduced Dietary fat	Anti-inflammatory Diet	Reduced Protein	Gluten Free Diet	Modified Carbohydrate	Low FODMAP Trial	High Soluble Fibre	ReducedFibre	Low Chemical Diet	Low Food Additives	Probiotic Supplements	Prebiotic Supplements	Complementary Therapies	Reduced Caffeine
Predisposing or risk factors for FD
Suspected duodenal microbiota alterations																
Small intestinal bacterial overgrowth				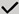			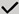	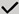								
Suspected immune or allergy-like response																
Intestinal permeability			. 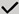	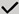	? L-glutamine	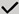						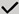	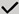	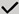		
Bile acid involvement																
FD symptom-related
Delayed gastric emptying and/or impaired accommodation					. 					Soluble					Ginger Iberogast	
Early satiety	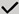	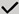	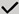		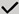					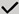						
Wheat-induced symptoms																
Pain	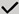						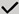	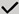				Polyols			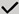	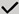
Co-presenting extraintestinal symptoms (Atopy, migraine)																
Post prandial distress																


 dietary management approach suited to symptom or risk factor. ? dietary management option to consider (very limited evidence or only suitable for a subset of patients).

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
