# Peer review of "Mechanisms of Food-Induced Symptom Induction and Dietary Management in Functional Dyspepsia"

_nutrients, 2021, doi:10.3390/nu13041109_

Round 1
Reviewer 1 Report
This is a very appreciable paper, fully describing any aspect of functional dyspepsia. I have appreciated the attention in the description of food-induced mechanisms of functional dyspepsia, even if some information is repeated a little too many times between paragraphs.
I want to report some notifications:
-Title: since the aim of the Paper is to ‘..collate the possible mechanisms for eating-induced and food-related symptoms and use the available evidence to develop and present a staged process for dietary management of FD’ the Authors could hypothesize to modify the title in order to introduce also the concept of describing the mechanism for eating and food-induced symptoms of FD.
-The Paper presents too many acronyms making hard the text comprehension. Please avoid some acronyms presented only a few times, such as SI, SCD, MAM
-The References style should be modified according to the journal (see Author’s Guidelines)
-Section number 1‘Introduction’ has an additional initial point
-In the ‘Introduction’ section, when explaining differential diagnosis with possible diseases associated with dyspepsia (lines 31-32), the Authors should include also atrophic gastritis.
-In Lines 384-385 Authors refer to Table 1 and Figure 1 not presented in the paper. Tables and Figures should be added in the template in order to evaluate them.
-Title of section number 7 should be modified since there are too many ‘and’; a comma is suggested instead of the first ‘and’.
-In section number 9 the expert opinion paragraphs start. This should be clearly stated in the section’s title, in addition to the text.
Moreover, in the same section, the sub-paragraphs are not well defined since the titles are not evidenced and numbered (line 820, line 847, line 887, line 917, line 937, line 953, line 966). Please modify.
Reviewer 2 Report
Dear Editor, I would like to thank you for giving the opportunity to review the manuscript by Duncanson et al. Authors are to be commended for putting together a concise and well-written manuscript on a very interesting yet underrated topic. I would only suggest some minor trimming of the Introduction section since it is way too long containing many redundant information.
Round 2
Reviewer 1 Report
Thanks for your modifications.